# GPCR-specific autoantibody signatures are associated with physiological and pathological immune homeostasis

Otavio Cabral-Marques et al.[#]

Autoantibodies have been associated with autoimmune diseases. However, studies have identified autoantibodies in healthy donors (HD) who do not develop autoimmune disorders. Here we provide evidence of a network of immunoglobulin G (IgG) autoantibodies targeting G protein-coupled receptors (GPCR) in HD compared to patients with systemic sclerosis, Alzheimer's disease, and ovarian cancer. Sex, age and pathological conditions affect auto-antibody correlation and hierarchical clustering signatures, yet many of the correlations are shared across all groups, indicating alterations to homeostasis. Furthermore, we identify relationships between autoantibodies targeting structurally and functionally related mole-cules, such as vascular, neuronal or chemokine receptors. Finally, autoantibodies targeting the endothelin receptor type A (EDNRA) exhibit chemotactic activity, as demonstrated by neutrophil migration toward HD-IgG in an EDNRA-dependent manner and in the direction of IgG from EDNRA-immunized mice. Our data characterizing the in vivo signatures of anti-GPCR autoantibodies thus suggest that they are a physiological part of the immune system.

---

More than a century after the immunologist Paul Ehrlich proposed his theory of horror autotoxicus based on the concept that immunized animals did not produce autoantibodies (aab) in response to their own blood or blood from their own species[1], a paradigm persists linking aab to the development of autoimmune diseases[2]. However, aab have been found in healthy donors (HD) at preclinical stages and even in those who never develop autoimmune disorders[3,4]. Furthermore, beneficial, naturally occurring aab that show protective effects against the development of immune-mediated diseases, such as type 1 diabetes and psoriasis, have recently challenged the aforementioned paradigm[5]. The most common theories proposed to explain aab production are based on molecular mimicry and immune dysregulation[4,6]. However, these theories mainly aim to integrate the mechanisms of aab production with the commonly accepted paradigm that associates aab with autoimmune diseases. Thus, they are unable to fully explain the occurrence of self-reactive B cells in mice and humans[7] and the production of immunogobulin G (IgG) aab that are naturally present in sera from HD. The generation of natural aab shares a common ontogeny with that of conventional antibodies, as both depend on the presentation of stimulatory antigens by dendritic cells to T and B lymphocytes[4,6]. We hypothesize that, similar to the dysregulation of any biological process, such as the imbalance of cytokine synthesis by T helper (Th) cells in several pathological conditions[8], the dysregulation of aab production and function may lead to autoimmune diseases. Thus, we suspect that the homeostasis of aab relationships, which are possibly a physiological part of our immune system, may break down, causing autoimmune disease.

We and other research groups have previously reported the existence of functional aab targeting G protein-coupled receptors (GPCRs) in patients with rheumatic diseases[9]. GPCRs are the largest superfamily of integral membrane proteins in humans[10]. GPCRs play an essential role in vertebrate physiology by sensing the external environment of a cell and responding to a variety of physiological stimuli[11]. For instance, GPCRs coordinate the cellular behavior involved in host immune responses[12] by acting as chemokine receptors, thus functioning as pivotal regulators of cell migration and cell trafficking throughout the body. In this context, GPCRs have been shown to interact with other essential physiological molecules by, for instance, cross-communication with growth factors and growth factor receptors by generating transactivation signals that contribute to the control of cell migration[13].

Here, our aim is to employ a stepwise, integrated systemic immunology approach to extensively characterize the correlation signatures of anti-GPCR aab across multiple chronic diseases and in a large cohort of healthy humans. We find a network of aab in sera from HD that target GPCRs. These aab also correlate with other aab directed against growth factors, growth factor receptors, and signaling molecules. The aab signatures are dependent on factors such as age, gender, and pathological conditions and have both shared and divergent components in a wide range of diseases (systemic lupus erythematosus or SLE, granulomatosis with polyangiitis or GPA, rheumatoid arthritis or RA, systemic sclerosis or SSc, ovarian cancer or OC, and Alzheimer's disease or AD). Specifically, we also found that anti-GPCR aab targeting human endothelin receptor type A (EDNRA) regulate neutrophil migration. Our data provide support to the concept that anti-GPCR aab are natural components of human biology. When the production of anti-GPCR aab becomes dysregulated, they may trigger the development of autoimmune diseases.

## Results

**Disease-specific signatures of aab targeting GPCRs.** Since both elevated and decreased concentrations of aab have been associated with the development of immune-mediated diseases[14–24], we suspected that anti-GPCR aab are an intrinsic part of the immune system after observing altered levels of multiple anti-GPCR aab in sera from patients with different autoimmune diseases, such as SLE, SSc, GPA, and RA, compared with healthy subjects (Fig. 1a–c, Supplementary Fig. 1). Our analyses revealed disease-specific signatures of aab concentrations compared with those of healthy individuals. SLE patients displayed increased concentrations of all 10 anti-GPCR aab tested. While the levels of some anti-GPCR aab were similar between the HD and disease groups, SSc and RA patients exhibited both elevated and decreased aab concentrations, and patients with GPA frequently demonstrated lower anti-GPCR aab concentrations compared with HD. We also observed that aab directed against structurally and functionally related molecules, such as anti-angiotensin II receptor type 1 or AGT1R and EDNRA (Fig. 1d–i), cholinergic receptor muscarinic (CHRM) 1–4 (Fig. 1j), protease-activated receptors (coagulation factor II thrombin receptor or F2R and the coagulation factor II thrombin receptor-like 1 or F2RL1) (Fig. 1k), and chemokine (C-X-C) motif receptor 3 (CXCR3) and 4 (Fig. 1l), correlated strongly in HD. In this context, the association between aab targeting CXCR3 and CXCR4 remained stable despite the presence of autoimmune diseases. However, disease-specific changes among the other aab were identified. While only SLE abrogated the normal interconnection between anti-AGTR1 and anti-EDNRA (Fig. 1e–i), the correlation among anti-CHRMs was reduced in SLE and GPA, whereas the partnership of aab directed against F2R and FRL1 was abolished in SLE, SSc, and GPA.

**Network-based view of aab targeting GPCRs.** The aforementioned data suggested physiological relationships among aab. To gain better insights into this field of investigation, we expanded our study to include the analysis of HD sera for correlation networks among anti-GPCR aab and other aab groups (e.g., growth factors and growth factor receptors, as well as signaling and neuronal molecules; Supplementary Tables 1 and 2). As previously shown for a few types of aab[15], we observed that HD aab targeting GPCRs strongly correlated with each other and with aab targeting growth factors and growth factor receptors (Fig. 2a). While gender and age slightly modified the relationship between sets of aab from HD (Fig. 2c–f), the results were markedly different in sera from patients with SSc (Supplementary Table 2, cohort 1; Fig. 2b), a disease for which anti-GPCR aab-associated immunopathological mechanisms have been extensively investigated[9,19]. For instance, aab correlations changed, including those between aab targeting AGTR1, epidermal growth factor receptor (EGFR), and vascular endothelial growth receptors (FLT1 and KDR) and the other aab.

To identify changes in the aab relationships of patients with non-autoimmune diseases, we analyzed aab correlations in conditions that are known to be associated with gender and age but of clearly different etiologies, such as OC and AD. In OC, the interconnections among aab (Supplementary Table 2, aab dataset 2) exhibited only a slight difference in comparison to HD (Fig. 3a, b). In patients suffering from AD, the global interconnection among aab-targeting GPCRs, growth factors, growth factor receptors, and neuronal molecules (Supplementary Table 2, aab dataset 3), which have been previously associated with cognitive dysfunction and depression[25], were only marginally different (Fig. 3c, d). Of note, aab targeting neuronal receptors (dopamine, adrenergic, muscarinic, and

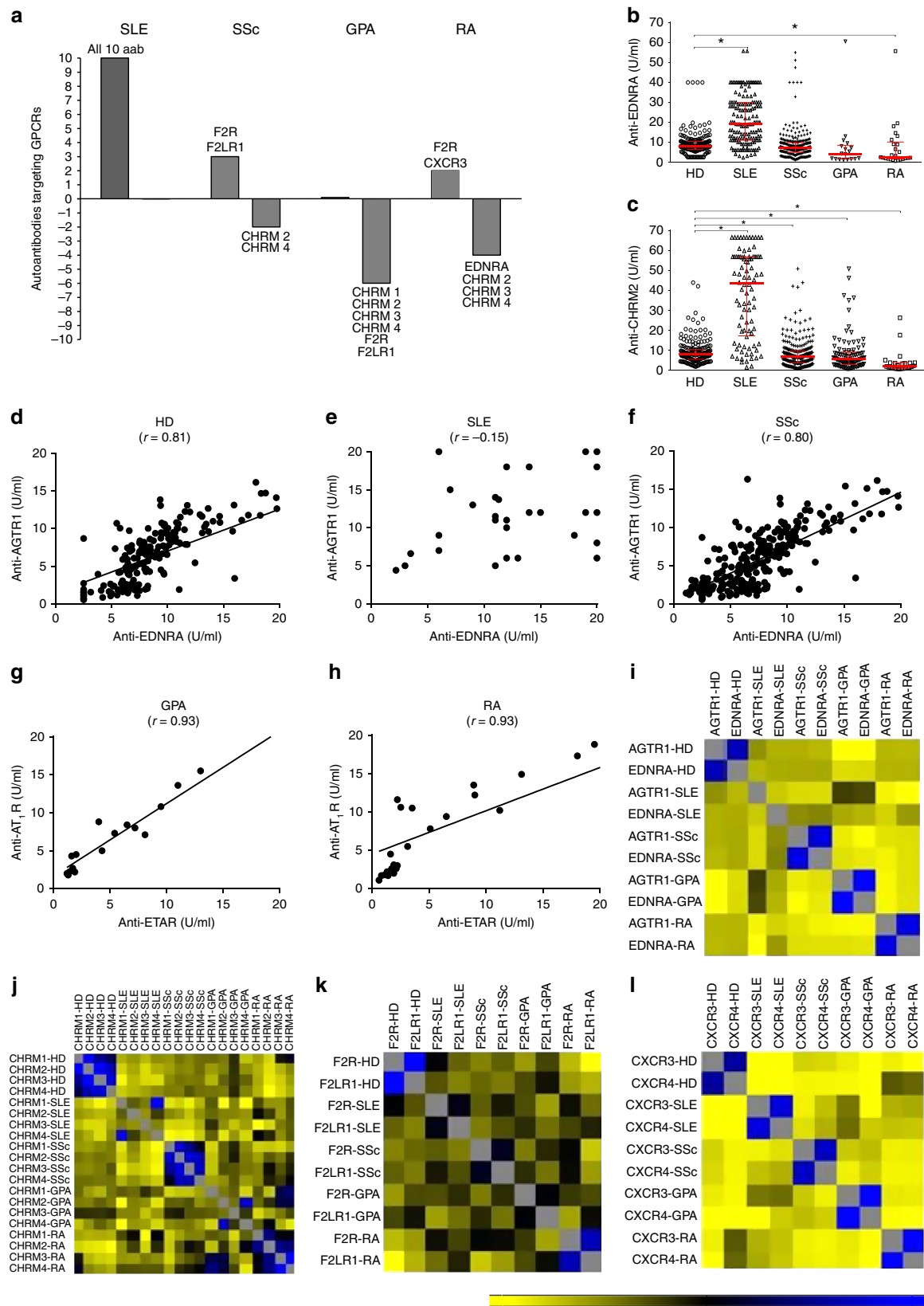

serotonin receptors) showed strong intragroup relationships in both HD and patients with AD, further indicating the presence of physiologically related aab. While aab concentrations and their associations with clinical data from the AD cohort have been previously described[25], detailed information regarding aab

levels in sera from SSc and OC patients and correlations with clinical characteristics are being prepared.

We suspected that the alterations in aab networks described above, which were more dramatic in sera from SSc patients, could be an effect of changes in the distribution pattern of aab

**Fig. 1** Relationships among autoantibodies in health and autoimmune diseases. **a** The graphic summarizes anti-GPCR autoantibodies (aab) in healthy donors (HD), which showed significantly increased or decreased concentrations when compared to those in the disease cohorts (systemic lupus erythematosus or SLE, systemic sclerosis or SSc, granulomatosis with polyangiitis or GPA, and rheumatoid arthritis or RA). Further details are shown in Supplementary Figure 1. The x-axis represents healthy controls. Graphics display concentrations of aab directed against **b** EDNRA and **c** CHRM2. The median with interquartile range is shown in red. *$p \le 0.05$ (Mann–Whitney test). Linear regression graphics exhibit the correlation between anti-EDNRA and anti-AGTR1 aab in sera from **d** HD, **e** SLE, **f** SSc, **g** GPA, and **h** RA. Heatmaps of aab vs. aab correlations demonstrate the spectrum of relationships among aab targeting **i** EDNRA and AGTR1; **j** CHRMs; **k** F2R and FLR1; and **l** CXCR3 and CXCR4 (for nomenclature, see Supplementary Table 2, aab dataset 1). The bar ranging from yellow to blue (−0.3 to 1) represents negative to positive correlations, respectively. In the correlation matrix, each small square represents a pairwise correlation between aab, as exemplified by **d**–**h**. The correlation matrices used to perform the hierarchical correlograms shown in Fig. 1i–l are provided as source data

concentrations across the different cohorts of HD and disease categories. To test this possibility, we calculated the Gini coefficient of inequality with bootstrap confidence intervals, which is the most popular metric for operationalizing inequality, defined by the mean of the absolute difference in all pairs of individuals for some measurements[26]. This approach revealed that the changes observed in aab relationships, as shown by the circular plots (Fig. 2 and Fig. 3) according to gender, age, and disease, reflected the distribution pattern of aab concentrations across the different investigated groups (Fig. 4). In accordance with the Gini index approach, linear discriminant analysis (LDA) of aab across the HD and disease groups highly discriminated HD and SSc patients when compared to the moderate differences between HD vs. OC and HD vs. AD (Supplementary Figure 2).

**Hierarchical clustering signatures of anti-GPCR aab.** To further characterize the relationships among aab, we performed hierarchical clustering analysis of aab correlations. For comparison, Fig. 5 shows nonclustering heatmaps of the correlations among all three aab datasets (Supplementary Table 2) investigated in the HD and disease cohorts. The hierarchical clustering analysis of aab correlations revealed several clusters (modules) of aab in sera from HD (Fig. 6). Dendrograms of the correlation matrices exhibited a strong proximity of aab targeting molecules that are closely related in terms of structure and function, such as AGTR1/EDNRA, CXCR3/CXCR4, CHRMs, and β-adrenergic receptors (ADRB), as well as between growth factors and growth factor receptors. Aab from healthy females and males below and above 65 years of age, the latter age group representing the life period most associated with increased generation of natural aab[3], formed major cluster modules (a term previously well characterized in the context of correlation network analysis[27]). We found several positive and negative clusters of correlations between various aab groups, supporting the modification of aab relationships by age and gender. In the presence of SSc (Fig. 6), AD, and OC (Supplementary Figure 3a–b), we observed disease-specific formation of aab clusters when compared to the HD subgroups. However, markedly changed aab hierarchical clustering was only observed when comparing aab from the SSc cohort (composed mainly of females, mean age 56.9 ± 13 years) to HD. Aab from the SSc and OC cohorts were located in the bottom right corner of the correlogram matrix near the clusters of aab from HD females above and below 65 years of age, respectively. Due to the small number of males aged >65 years in HD cohort 3, which was a limitation of our study, we did not perform subgroup analyses of HD cohort 3 according to age in the correlogram matrix of the associations among aab from AD patients. Hierarchical clustering analysis of aab in HD alone (Supplementary Table 1, aab dataset 1) and from HD compared with patients with SSc, OC, or AD when not subgrouped by gender and age are shown in Supplementary Figure 3c–e. Taken together, these findings suggest that gender, age, and pathological

condition influence the hierarchical clustering signatures formed by aab.

**Relationships among anti-GPCR aab.** To integrate the analysis performed using three cohorts of HD and patients diagnosed with different diseases and using different aab datasets, we performed a multistudy factor analysis (MSFA)[28]. In agreement with our hypothesis that aab acting under normal physiological conditions may become dysregulated under the influence of different variables, the MSFA revealed the presence of common and specific latent factors when comparing HD to patients with selected diseases (Fig. 7). This finding implies that the specific factors we observed in the HD and disease cohorts are involved in particular physiological and pathological mechanisms, respectively. In contrast, the common (shared) factors suggest physiological functions that are regulated by aab but not affected by the disease state.

Next, since an interactome database for aab is currently not available, we reverse-engineered aab functions through in silico evaluation of aab target interactions to gain insights into the dynamics of a putative aab network. Network mapping of aab targets (Supplementary Table 2, dataset 1) displayed multiple associations among GPCRs, growth factors, and growth receptors, suggesting an enriched network (Supplementary Figure 4a) in which EDNRA plays a central role. Enrichment gene ontology (GO) analysis indicated the regulation of cell migration (GO:0030334) as a strongly significant function among the biological processes involved in this network of aab targets (Supplementary Figure 4b).

**Chemotactic activity of anti-EDNRA aab.** Based on our in silico analysis, we assessed the effect of IgG from HD (HD-IgG) without and with EDNRA inhibition on the migration of neutrophils before testing the hypothetical chemotactic activity of anti-EDNRA aab. Neutrophils are the most abundant peripheral blood-circulating leukocytes and the first white cells to invade sites of inflammation, that is, during infection or following tissue damage[29]. Neutrophils, which express EDNRA (Fig. 8a and Supplementary Figure 5), migrated toward HD-IgG in an EDNRA-dependent manner (Fig. 8b). The requirement of EDNRA for HD-IgG-induced migration was confirmed using Colo357 cells, as shown in Supplementary Figure 6. The HD-IgG-mediated chemotactic activity was mainly induced by the Fab fragment of IgG (Fig. 8c), although the Fc fragment showed weak chemotactic properties as well. Notably, the effect of EDNRA inhibition on cell migration by sitaxsentan was not due to cytotoxic effects since neutrophils in vitro did not display signs of apoptosis or necrosis when EDNRA was inhibited by sitaxsentan (Supplementary Figure 7). HD-IgG did not affect other physiological functions, such as the neutrophil or monocyte oxidative burst or lymphocyte proliferation (Supplementary Figure 8; both performed as previously

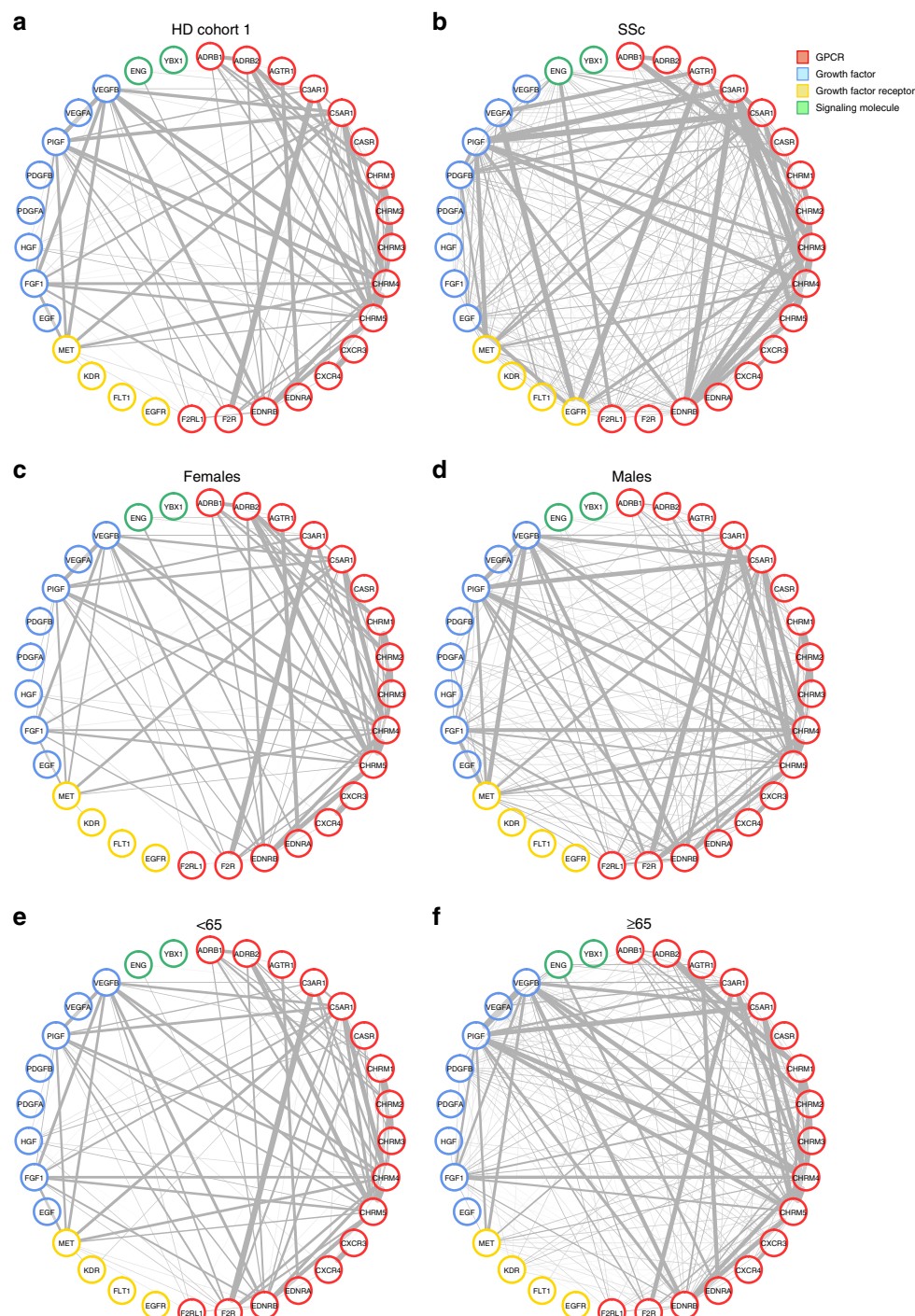

**Fig. 2** Effects of gender, age, and systemic sclerosis on autoantibody correlations. We analyzed the relationships among the different autoantibodies (aab) in sera from healthy donors (HD) and the effects of gender and age, as shown by circular networks based on Spearman's rank correlation coefficients for aab. Circle plots show the correlation matrix of aab comparing each condition: **a** all HD evaluated, **b** patients with systemic sclerosis (SSc; Supplementary Table 1, cohort 1; Supplementary Table 2, aab dataset 1); subgroups of HD (cohort 1) analyzed according to **c**, **d** gender and **e**, **f** age (< and ≥65 years)

described[30,31]). In addition, the HD-IgG used was endotoxin-free, as demonstrated by endotoxin testing (PTS cartridge <0.05 EU/ml). Considering the central role of interleukin-8 (IL-8) as an important chemokine that regulates neutrophil migration[29], we were able to show that HD-IgG triggered IL-8 production by peripheral blood mononuclear cells (PBMCs) (Fig. 8d), which also express EDNRA[32]. The spontaneous IL-8 synthesis by PBMCs strongly correlated with EDNRA expression (Fig. 8e). These findings suggest that HD-IgG controls the

trafficking of neutrophils directly via chemotactic mechanisms and indirectly by triggering IL-8 production.

To investigate a possible chemotactic effect of anti-EDNRA aab on cell migration, C57BL/6 mice were immunized with membrane extracts of Chinese hamster ovary (CHO) cells overexpressing human recombinant EDNRA (Fig. 8f), a molecule that is highly conserved between humans and mice, as well as across several other species (Supplementary Figure 9). EDNRA-immunized mice produced high concentrations of aab targeting

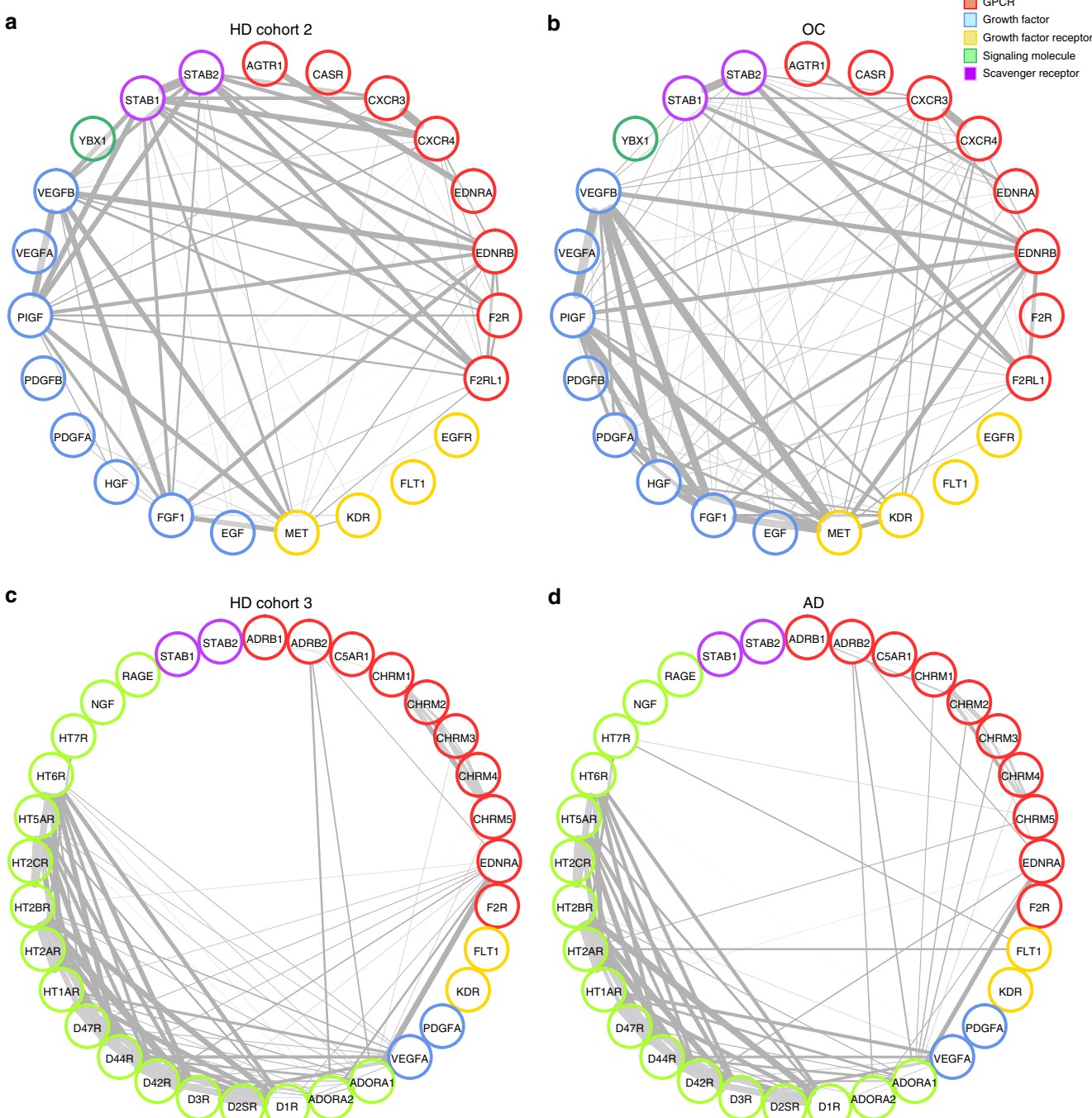

**Fig. 3** Effects of ovarian cancer and Alzheimer's disease on autoantibody correlations. Graphics display the comparisons of **a** healthy donors (HD) and patients with **b** ovarian cancer (OC; Supplementary Table 1, cohort 2; Supplementary Table 2, aab dataset 2) and **c** HD and patients with **d** Alzheimer's disease (AD; Supplementary Table 1, cohort 3; Supplementary Table 2, aab dataset 3). The nodes in the graphs represent variables (each aab), and a line between two nodes indicates the Spearman's rank correlation coefficient. The line width indicates the strength of the association, with stronger correlations indicated by thicker lines. Only correlations >0.6 are shown. Multiple connections of nodes indicate clustering

EDNRA (Fig. 8g). Human neutrophils migrated toward IgG from both control and EDNRA-immunized mice. However, IgG from the latter group showed stronger chemotactic activity and induced the formation of neutrophil aggregates when compared to IgG from non-immunized mice (Fig. 8g).

## Discussion

With the discovery of anti-GPCR aab in patients with allergic rhinitis and asthma[33] and the recognition that they play a role in

the pathogenesis of several disorders, such as rheumatic diseases[9], the physiological relevance of aab targeting GPCRs found in the sera of HD[9,34] remains poorly understood. While the function of some of the aab reported herein and their value as disease biomarkers are still under investigation[16,24], other anti-GPCR aab, such as those targeting EDNRA and AGTR1, were previously well described. These studies utilized purely analytical approaches and considered EDNRA and AGTR1 to be pathological agents that act on different molecular targets by triggering intracellular GPCR signaling[9,15,35]. The network-based analyses carried out in our

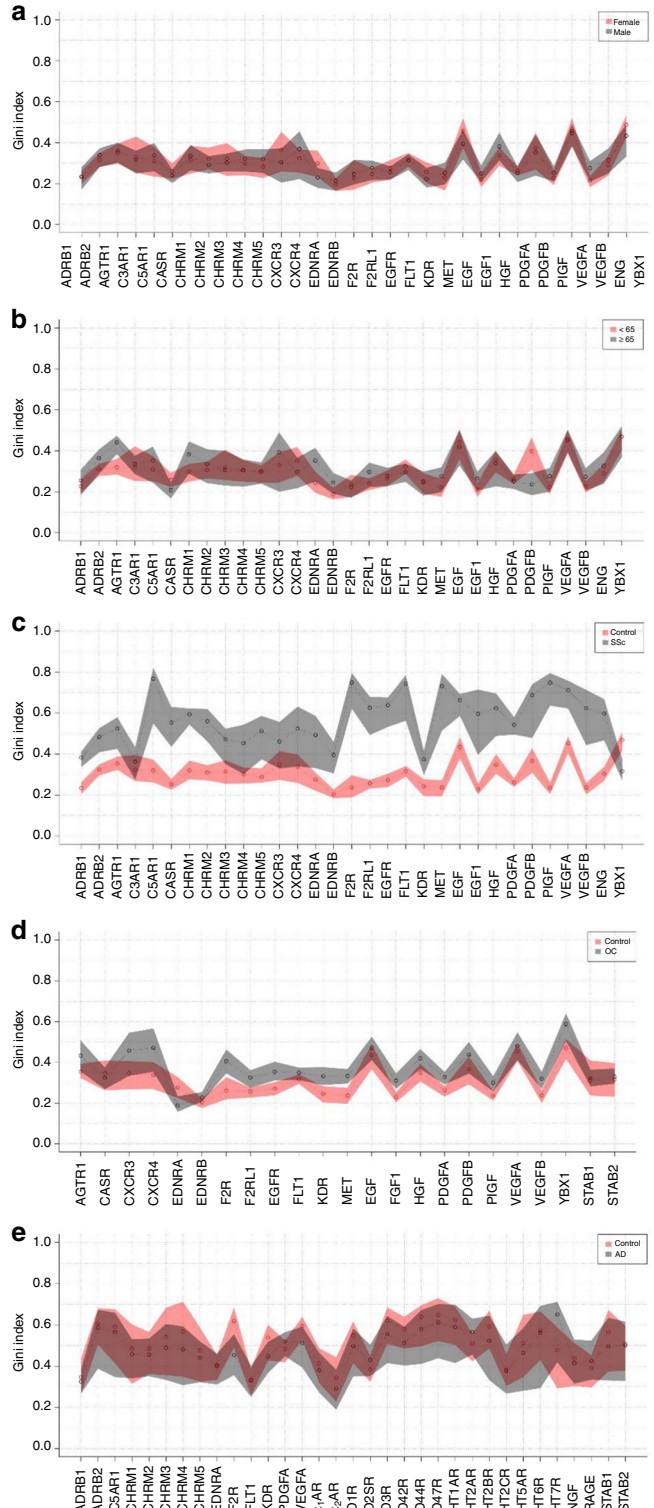

**Fig. 4** Autoantibody relationships reflect their concentration distribution patterns. Gini index confidence intervals were obtained by bootstrap analysis. The red and gray shadows represent confidence intervals, and each small circle indicates the Gini index value. The graphics exhibit comparisons of **a** HD females and males and **b** HD above and below 65 years of age (Supplementary Table 1, cohort 1; Supplementary Table 2, aab dataset 1) and comparisons between **c** HD and patients with systemic sclerosis (SSc, Supplementary Table 1, cohort 1; Supplementary Table 2, aab dataset 1, **d** HD and patients with ovarian cancer (OC, Supplementary Table 1, cohort 2; Supplementary Table 2, aab dataset 2) or **e** HD and patients with Alzheimer's disease (AD, Supplementary Table 1, cohort 3; Supplementary Table 2, aab dataset 3)

our recently developed novel SSc mouse model, which is based on increasing the serum concentration of anti-AGT1R aab by immunizing mice with human AGTR1 (manuscript submitted). Therefore, our observations support the hypothesis that anti-GPCR aab are natural components of the immune system and may become dysregulated, triggering the development of auto-immune diseases. This assumption is in accordance with the emerging observation of the role of the immune system in homeostasis beyond host defense[36–40].

Here, we determined the correlation signatures of aab in health and disease based on enzyme-linked immunosorbent assay (ELISA), an approach that is widely used to determine the presence of aab that target GPCRs[9]. Although this method is well established, there are limitations. The avidity and affinity of aab to their target are not measured. Of note, aab avidity and affinity, as well as aab isotype, can change the outcome of antibody–antigen interactions and relevant biological processes following binding. As such, some biological determinants of harboring GPCR aab have not been measured for each aab in our study, which may be potentially relevant for all patient groups, including patients with autoimmune diseases. Consequently, it will be important to investigate the characteristics of the aab described above in HD to determine their influence on aab physiology. For instance, low concentrations of aab could be compensated by high binding affinity and vice versa. We are also expanding our current findings by analyzing specific epitopes. Furthermore, we are establishing experimental immunization models to better understand the pathophysiology of aab targeting GPCRs. Thus far, immunizations of mice with human AGTR1 and EDNRA have been successful, and reports will be published in detail elsewhere. For instance, immunization with AGTR1, a highly conserved receptor in humans and mice, increases functional anti-AGTR1 ab levels. Such experimental immunization also induces pathological SSc features, including interstitial lung disease and skin fibrosis. Passive immune transfer studies would be a relevant next step to determine whether anti-AGTR1 indeed has pathogenic effects with respect to SSc. The presence of circulating aab in HD that target self-antigens, such as GPCRs, which are conserved among species[3] and form network signatures, expands our view of the immune system. The prevailing concept of a functional immune system has focused on its role as a guardian against invading pathogens. An evolving shift backing off from this paradigm is derived from the notion that the immune system has evolved not only to protect the host from pathogens but also to maintain the immunological homeostasis of the organism. As a consequence of internal tissue damage following infectious or noninfectious inflammation, this homeostasis can break down[36–40]. The current understanding of aab function is evolving and includes the notion that aab provide and maintain homeostatic functions through the binding of cellular

investigation revealed distinct signatures of anti-GPCR aab in HD that are influenced by age, gender, and various diseases. In addition, anti-GPCR aab from HD are possibly able to act as regulators of GPCR-mediated mechanisms, as evidenced by the chemotactic effects of human IgG and anti-EDNRA antibodies on cell migration. Therefore, together with the previously demonstrated effects of anti-GPCR aab on GPCR signaling[15,35], our findings suggest that aab targeting GPCR are newly recognized components of human biology. These data are in accordance with

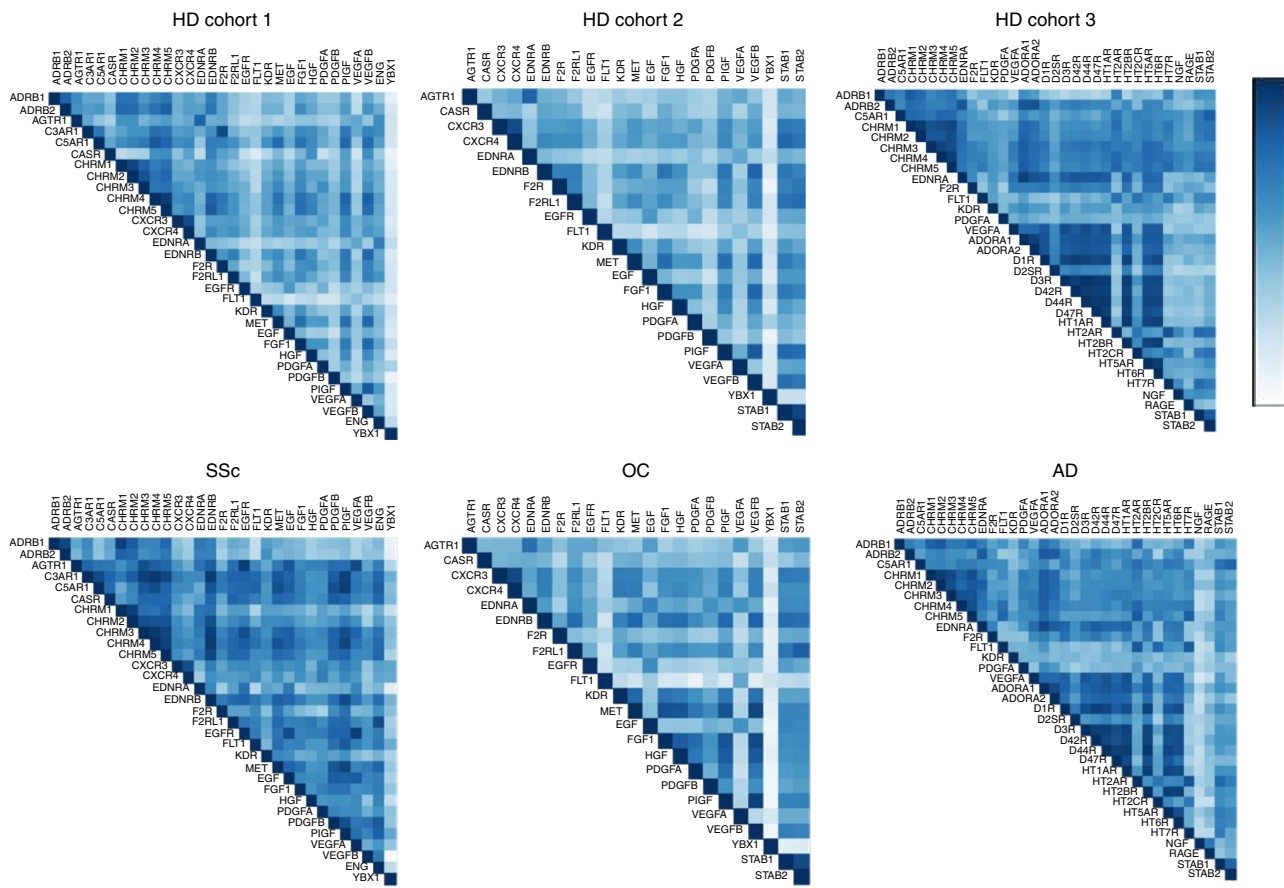

**Fig. 5** Heatmaps of autoantibody correlations. The images show Spearman's correlation of autoantibody datasets (Supplementary Table 2) **a** 1, **b** 2, and **c** 3 in sera from healthy donor (HD) cohorts 1–3 and patients (SSc systemic sclerosis, OC ovarian cancer, and AD Alzheimer's disease), respectively (Supplementary Table 1). The color scale bar (0 to 1) corresponds to weak and strong correlations, respectively

antigens and consequently contribute to the clearance of apoptotic cells[41]. Aab production could be modulated by changing particular conditions, such as increasing or decreasing receptor expression, as demonstrated by injection with CHO cells over-expressing human EDNRA, thereby inducing anti-EDNRA antibody synthesis. Consequently, the production and effect of these aab would depend on the level of receptor expression in specific tissues. Thus, modulating receptor expression for target epitopes on the cell membrane by aab-mediated endocytosis could provide a mechanism for maintaining aab-mediated cellular homeostasis. In addition, GPCR and growth factor receptors are not only expressed by immune cells and different body tissues, they are also present in extracellular vesicles, such as exosomes, which are implicated in numerous pathologies, including cancer and inflammatory diseases[42]. Therefore, exosomes could represent another important physiological source for the stimulation of anti-GPCR aab production that remains to be investigated.

From an evolutionary perspective, our findings are consistent with the phenomenon that autoreactive B lymphocytes have not been excluded by natural selection and can remain functionally active[5,7,43–46], producing aab that are ubiquitously present and evolutionarily conserved[3,47]. This logic challenges the paradigm that links aab exclusively with the triggering of autoimmunity. Instead, a novel concept arises that considers anti-GPCR aab as an integral part of the immune repertoire, the function of which may broadly affect several biological mechanisms by targeting GPCRs. The dysregulation of anti-GPCR aab relationships following uncontrolled tissue injuries in some pathological

conditions opens opportunities for new investigations in auto-immune disease. Moreover, it will also be important in the future to determine how anti-GPCR networks and signatures interact with drugs, as GPCRs comprise almost one-third of all current drug targets in clinical medicine[48].

In conclusion, the presence of a physiological network of aab directed against GPCRs in HD opens new avenues for the identification of new homeostatic mechanisms regulated by aab. In this context, it is reasonable to suggest that physiological aab against GPCRs[9], such as functional aab targeting EDNRA[15], and other functional aab, such as those targeting growth factor receptors including anti-PDGFR antibodies[49] previously characterized in patients with SSc, could also regulate a myriad of biological mechanisms in synergism with endogenous ligands[50,51]. These possibilities remain to be investigated in HD. Further evaluation of these possible aab functions will not only contribute to our understanding of the signaling systems involved but also of the mechanisms that lead to autoimmune diseases and subsequently the development of new and specific therapies.

## Methods
**Participants**. We first measured the concentration of aab directed against 10 different GPCR in sera from patients with different autoimmune diseases who underwent follow-up in the Department of Rheumatology at the University of Lübeck and at the University Hospital Charité in Berlin. Patients with SLE ($n = 249$), SSc ($n = 379$), GPA ($n = 128$), and RA ($n = 196$) were included and diagnosed according to the established criteria[52–55], and the concentrations of aab were compared to those of HD ($n = 197$). Next, a total of 491 HD and patients with SSc ($n = 84$), AD ($n = 91$), and OC ($n = 207$) (Supplementary Table 1, cohorts 1, 2, and 3) classified according to previously established criteria[55–57] participated in the

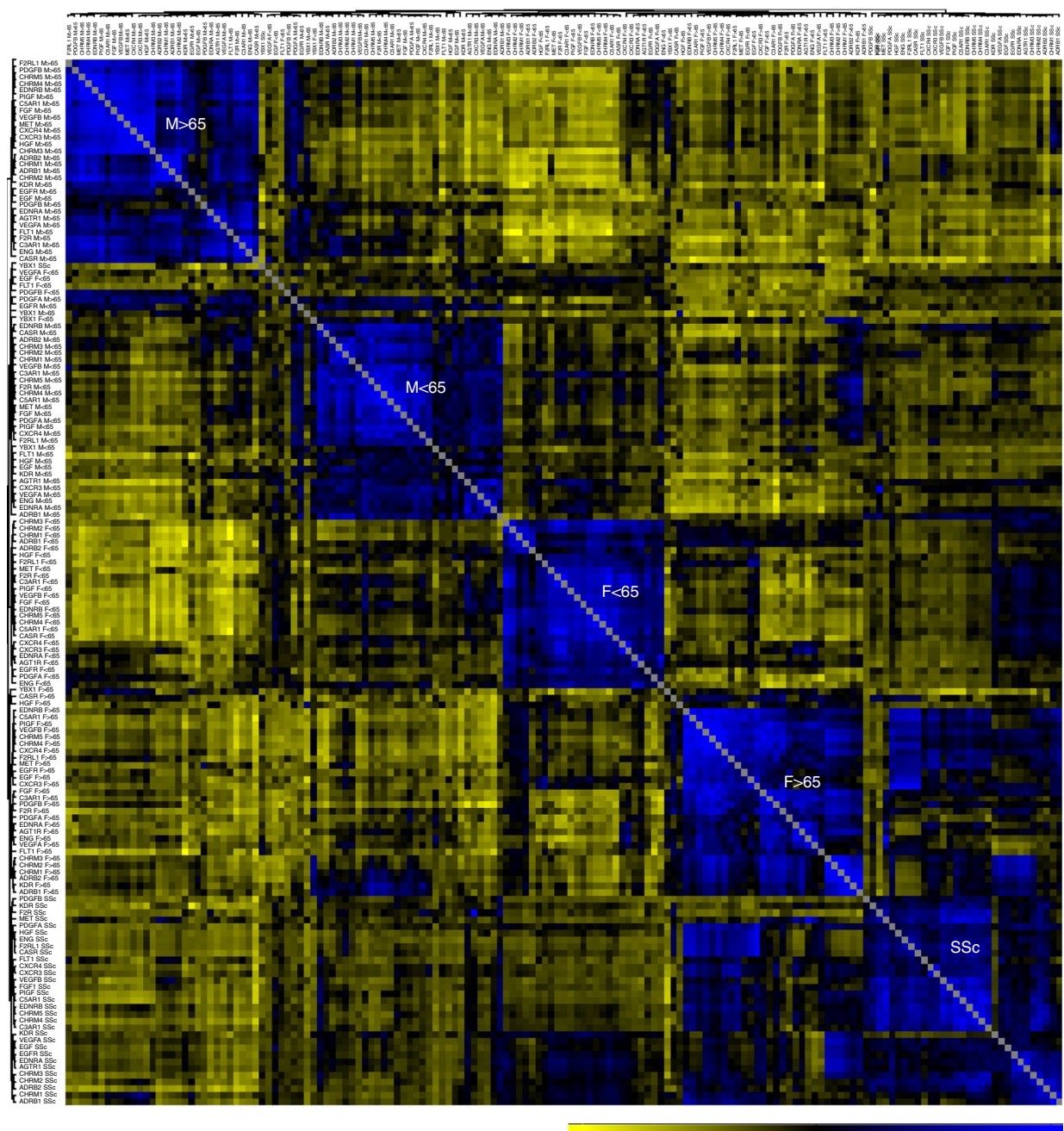

**Fig. 6** Hierarchical clustering of the autoantibody correlation signature. Correlogram matrix displays clusters (modules) of autoantibody (aab) correlations in all healthy donors (HD) according to gender and age (< and ≥65 years old) compared with patients with systemic sclerosis (SSc; Supplementary Table 1, cohort 1; Supplementary Table 2, aab dataset 1). Clusters of the correlations among aab are displayed in dendrograms on the top and side of the correlation matrix. The bar ranging from yellow to blue (−0.6 to 0.9) represents negative to positive correlations, respectively. In the heat map matrix, each small square represents the pairwise correlations between aab. The correlation matrix used to perform the hierarchical correlogram of SSc is provided as source data

study. Sera from SSc patients were provided by members of the Department of Rheumatology at the University of Lübeck and the University Hospital Charité in Berlin; the AD patients are being cared for by three participating Norwegian centers (Rogaland and Hordaland counties and the dementia study in Western Norway). Sera from OC patients were provided by Prof. Jalid Sehouli, Department of Gynecology, the University Hospital Charité in Berlin, corporate member of Freie Universität Berlin, Humboldt-Universität zu Berlin, and the Berlin Institute of Health. All HD and patients provided written consent to participate in the study, which was performed in accordance with the Declaration of Helsinki and approved by the ethics committees of the involved research centers.

**Aab detection by ELISA**. The methods to measure the aab have been previously described in detail[15,25]. Briefly, individual serum aab were assessed using

commercially available solid-phase sandwich ELISA Kits according to the manufacturer's instructions (all CellTrend GmbH, Luckenwalde, Germany). The aab concentrations were calculated as arbitrary units (U) by extrapolation from a standard curve of five standards ranging from 2.5 to 40 U/ml. The ELISAs were validated according to the Food and Drug Administration's Guidance for Industry: Bioanalytical Method Validation.

**Aab interaction network**. Statistical and bioinformatics analyses of the obtained aab data were performed using the free, open-source software package R[58]. The R packages are described in the Reporting Summary. We used circle plot diagrams to visualize the patterns of Spearman's rank correlation coefficients between aab and the Gini index coefficient[26] to assess the distribution patterns of aab concentrations. In addition, LDA of aab signatures was performed as previously described to

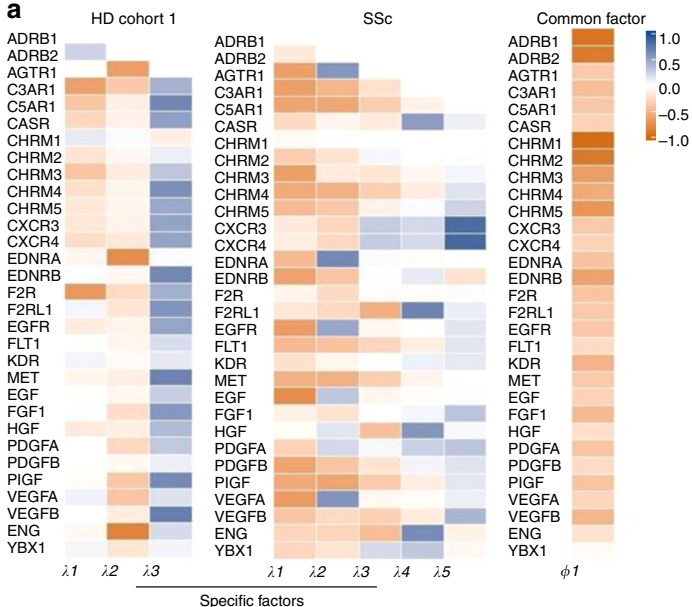

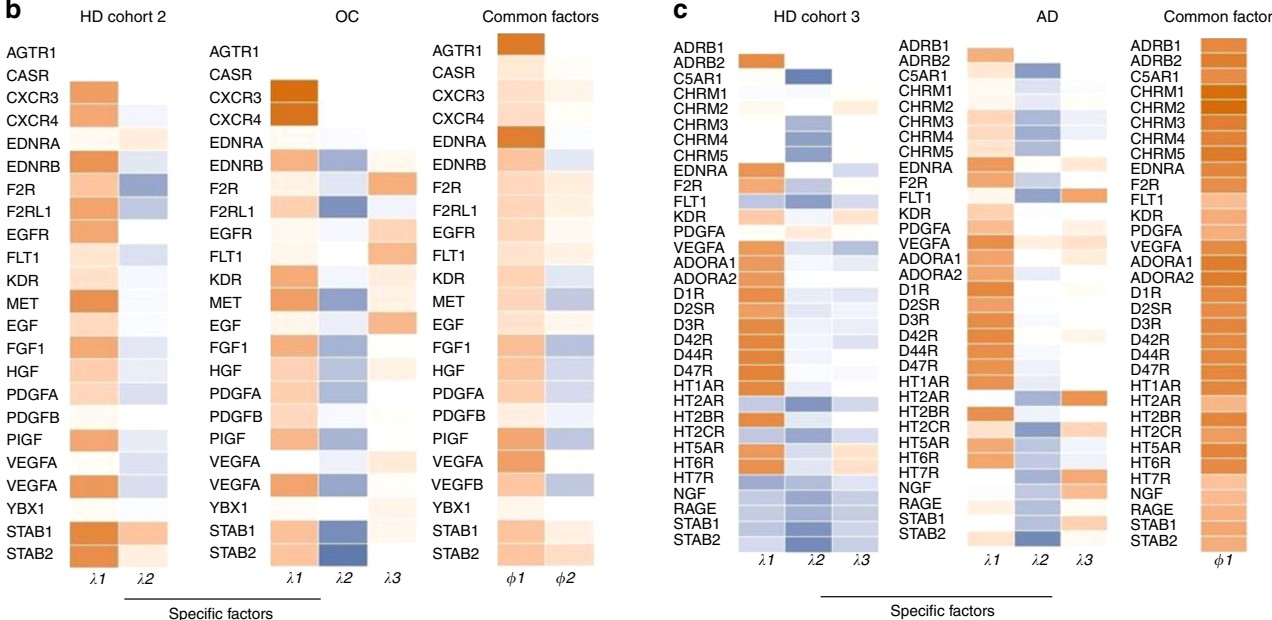

**Fig. 7** Multistudy factor analysis of autoantibodies. The multistudy factor analysis (MSFA) was performed to analyze autoantibodies (aab) from healthy donors (HD) compared with patients with **a** systemic sclerosis (SSc), **b** ovarian cancer (OC), and **c** Alzheimer's disease (AD). Supplementary Tables 1 and 2 provide further details about the HD and patient groups, as well as the aab datasets analyzed. The images are heatmaps of estimated factor loadings of common and specific latent factors. The color scale bar ranging from orange (−1 to 1) to blue corresponds to negative and positive factor loadings. Loadings close to −1 or 1 indicate aab that strongly influence factors in opposite directions

discriminate the global aab signature of HD and patient groups[59] as appropriately recommended to analyze immunological data[60]. MSFA (described in detail below), using a fast expectation conditional-maximization algorithm for the parameter estimate[28], was used to comprehend the correlation structure of the aab network. This approach is a generalized version of factor analysis that is able to handle multiple studies simultaneously to identify common and study-specific factors shared by different studies[28]. MSFA considers all data at once in an integrated approach, estimating parameters by maximum-likelihood analysis[61]. MSFA allowed the identification of specific and shared factors among aab from three different cohorts of HD (Supplementary Table 1) analyzed for three aab datasets (Supplementary Table 2) and in comparison to cohorts of patients with SSc, AD, and OC, respectively. Hierarchical clustering of Pearson's correlations of aab[62] was carried out using the Perseus software[63] (MaxQuant, v1.11, Martinsried, Germany) to assess the correlation signatures of aab. When indicated, HD were subdivided according to gender and age (< and ≥65 years of age), with the latter representing the life period most associated with increased dysregulation of natural antibody

production[3]. In addition, we reverse-engineered the functions of aab through the in silico evaluation of aab (aab dataset 1) target interactions using STRING[64] and GO analysis according to the Gene Ontology Consortium[65] to gain insight into the physiological roles[66] of aab. Enrichment analysis of the GO biological processes was considered significant when the false discovery rate/adjusted $p$ value was ≤0.05.

**IL-8 release.** HD-IgG (0.5 mg/ml) was used as a standard in all experiments as previously described[32]. In summary, after Fc-γ receptor blockade, $1 \times 10^6$ PBMCs isolated from heparinized blood as previously described[31,67] were activated by a pool of heterologous purified IgG from 10 different HD. After 20 h of culture in the absence or presence of the 100 μM EDNRA antagonist sitaxsentan (Pfizer Inc., New York, NY, USA), the supernatants were harvested and evaluated for the presence of IL-8 using ELISA according to the manufacturer's instructions (R&D Systems, Minneapolis, MN, USA).

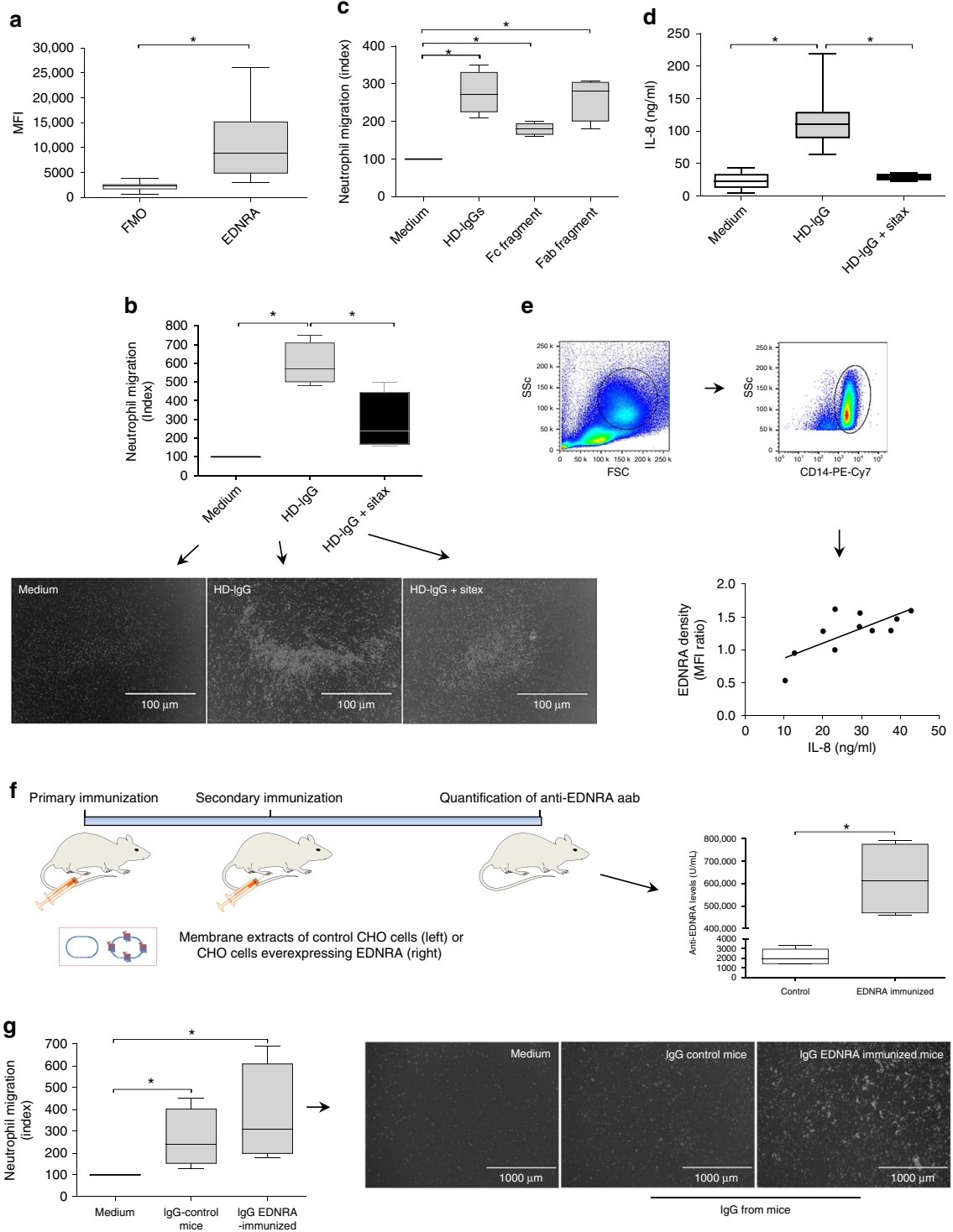

**Migration assays**. The capacity of antibodies to induce neutrophil chemotaxis was analyzed using the transwell migration assay (24-well plate, Corning Inc., Corning, NY, USA) as previously described[32]. For the analysis of HD-IgG-induced migration, neutrophils were isolated from heparinized blood by dextran sedimentation followed by Ficoll-Hypaque centrifugation as previously described[30]. The purity, based on CD15 expression, was always above 97%, as was the viability before the migration assays. After Fc-γ receptor blockade with Human TruStain FcX Receptor Blocking Solution (Biolegend, San Diego, CA, USA), the isolated neutrophils (0.2 × 10⁶) were transferred to the upper chamber, while RPMI containing total human IgG, Fc, and Fab fragments (Merck, Darmstadt, Germany) or IgG from EDNRA-

immunized and control mice (kindly provided by Prof. Xinhua Yu and Prof. Frank Petersen from the Research Center Borstel, Airway Research Center North, ARCN, Members of the German Center for Lung Research DZL, Borstel, Germany) were placed in the lower chamber of plates and incubated at 37 °C for 1 h. When indicated, neutrophils were incubated before chemotaxis assays for 1 h in the absence or presence of 100 µM of the EDNRA antagonist sitaxsentan. Migrated neutrophils present in the bottom of the transwell plates were transferred to a 96-well plate and counted (cells/µl) by flow cytometry using a CytoFLEX Flow Cytometer (Beckman Coulter, Indianapolis, IN, USA) by gating on neutrophils according to size (forward scatter, FSC) and granularity (side scatter, SSC) to

**Fig. 8** Effect of HD-IgG and anti-EDNRA autoantibodies on neutrophil migration. **a** Expression of endothelin receptor type A (EDNRA) by human neutrophils (n = 16). The fluorescence minus one control (FMO) was analyzed as shown in Supplementary Figure 5. **b** Neutrophil chemotaxis toward 0.5 mg/ml IgG from healthy donors (HD-IgG) in the presence or absence of the EDNRA antagonist sitaxsentan (sitax; n = 3). A representative image of neutrophils (white dots in the figure) on the bottom surface of transwell plates is shown. **c** Neutrophil migration toward intact human IgG, antigen-binding fragment (Fab), and the crystallized fragment (Fc) region. The results are representative of three independent experiments (n = 3). **d** HD-IgG-induced IL-8 production by peripheral blood mononuclear cells (PBMCs) in the absence (n = 13) or presence of sitax (n = 4). **e** IL-8 spontaneously released into the culture supernatants correlates with the level of EDNRA expression on CD14+ monocytes (n = 11). **f** Concentrations of anti-EDNRA autoantibodies (aab) in mouse sera were assessed after secondary immunization with membrane extracts from control Chinese hamster ovary (CHO) cells (n = 5) or CHO cells overexpressing human EDNRA (n = 4). Mouse images as well as syringe and membrane cartoons were adapted from Motifolio Drawing Toolkits (www.motifolio.com). EDNRA immunization was carried out by administration into footpads of 0.2 mg of membrane extracts prepared from CHO cells overexpressing human EDNRA (Celltrend, Germany). Three weeks after the primary immunization, mice were boosted with the same amount of membrane emulsified with incomplete Freund's adjuvant (IFA, Sigma-Aldrich, USA). In the control group, mice were treated with the same amount of membrane extract from untransfected CHO cells. Six weeks after the booster immunization, all mice were sacrificed for sample collection and quantification of anti-EDNRA aab. **g** The migration of human neutrophils (white dots in the figure) toward IgG from control and EDNRA-immunized mice (n = 3). Error bars denote SD. *p ≤ 0.05 (unpaired t test)

exclude cell debris. Images of the cells on the bottom surface of the transwell plates were obtained using a fluorescence microscope (EVOS FL Cell Imaging System, Oakwood, OH, USA). The migration index was calculated in relation to the spontaneous (medium) migration (index 100). In addition, cell migration toward HD-IgG in an EDNRA-dependent manner was further analyzed using a cell-based Oris™ migration assay with human pancreatic carcinoma Colo357 cells according to the manufacturer's instructions as previously described in detail[68]. Briefly, Colo357 cells were cultured and allowed to migrate (in the presence or absence of $10^{-4}$ M sitaxsentan) into the free space in the middle of the well for 48 h. The cells were fixed and stained with a DiffQuick Cell Staining Kit (Medion Diagnostics, Gräfelfing, Germany). After staining, an Oris™ detection mask was clipped to the bottom of the plate, and images were obtained with a blackfly camera on an Axioskop HBO 50 microscope. The migration area was determined by analyzing the migration images with the Fiji module of the ImageJ software.

**Analysis of EDNRA expression by flow cytometry.** Neutrophils were analyzed after red blood cell (RBC) lysis using a RBC lysis solution (Qiagen, Valencia, CA, USA) according to the manufacturer's instructions, and PBMCs were isolated as described above. The cells were stained for EDNRA expression using previously reported antibodies (Supplementary Table 3) and procedures[32,69]. Briefly, neutrophils and monocytes were gated according to size (FSC) and granularity (SSC) and based on the pattern of CD15 and CD14 expression (Supplementary Figure 8). For median fluorescence intensity (MFI) values, an isotype control (Supplementary Table 3) was used as described below to compensate for changes in the cytometry instrument sensitivity. Since the neutrophils and monocytes were analyzed in two different labs (University of Lübeck and Charité University Hospital, respectively), the latter cell population was analyzed on a MACS Quant Cytometer (Miltenyi Biotec) and the former using a CytoFLEX Flow Cytometer (Beckman Coulter, Indianapolis, IN, USA). The MFI for EDNRA expression was obtained using the FlowJo Software (Treestar Inc., Ashland, OR, USA). The CytoFLEX and MACS Quant Flow cytometers were calibrated using Cytoflex Daily QC Fluorospheres (Beckman Coulter, Indianapolis, IN, USA) and MACSQuant™ Calibration Beads (Miltenyi Biotec, Bergisch Gladbach, Germany), respectively, according to the manufacturer's instructions. MFI was determined by controlling the daily technical variability using a fluorochrome-conjugated isotype control antibody. Therefore, a subtraction procedure[70] was applied to obtain the EDNRA MFI density (MFI-EDNRA/MFI isotype).

## Data availability

A reporting summary for this article is available as a Supplementary Information file. The source data underlying Fig. 1b–l, Fig. 6, Fig. 8, and Supplementary Figures 3B–3C are provided as a Source Data file. Additional data that support the findings of this study are available from the corresponding author upon reasonable request. All R packages used in this manuscript are described in the Reporting Summary. The results of this study have been shared via Figshare and can be accessed using the following link: https://figshare.com/s/c24828b4a6260b2e5531.

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

## Acknowledgements

We thank all patients and HD for their participation in this study. We acknowledge Prof. Xinhua Yu and Prof. Frank Petersen from the Research Center Borstel, Airway Research Center North (ARCN), and Members of the German Center for Lung Research (DZL), Borstel, Germany, for providing sera from EDNRA-immunized and non-immunized mice. We thank the Mirjam Lichy Foundation for their support as well as the DFG (grant no. RI 1056-11/12) for financial support. We also thank Actelion Pharmaceutical GmbH, the Eppenauer/Gutzeit Foundation, the German Network of Systemic Sclerosis and the EUSTAR network for their support. This study was supported by the Charité University Hospital in Berlin and University Hospital of Schleswig-Holstein, Campus Lübeck.

## Author contributions

O.C.-M., G.R., H.H., A.M., L.M.G., and B.K.al-R. conceived the project and designed the study; O.C.M., H.H., G.M., S.P., A.M., J.R., J.G., F.G., and C.P. performed the experiments; O.C.-M., G.R., A.M., H.D.O., L.M.G., B.K.al-R., M.J.F.-C., L.F.S., J.M., and A.M. cowrote the manuscript and provided scientific insights; O.C.-M., A.M., R.deV., B.E.E., and L.M.G. performed the statistical and bioinformatics analyses; G.R., T.L., J.Y.H., S.K., S.S., L.F.S., S.A., R.D., D.N.M., I.B., J.S., T.T., C.S., A.S., P.R.M., M.L., D.D., and P.L. diagnosed, recruited. or followed-up the patients; and H.H., K.S.F., G.R., T.L., J.Y.H., S.K., S.S., L.F.S., S.A., R.D., D.N.M., I.B., J.S., T.T., C.S., A.S., P.R.M., M.L., D.D., and P.L. coordinated the serum collection and databank.

## Additional information

**Competing interests:** The authors declare that H.H. and K.S.F. are CellTrend managing directors and that Gabriela Riemekasten is an advisor of the company CellTrend and earned an honorarium for her advice between 2011 and 2015. The other authors declare no competing interests.

Otavio Cabral-Marques[1,2], Alexandre Marques[1,3], Lasse Melvær Giil[4], Roberta De Vito[5], Judith Rademacher[6,7], Jeannine Günther[8,9], Tanja Lange[1], Jens Y. Humrich[1], Sebastian Klapa[1], Susanne Schinke[1], Lena F. Schimke[1], Gabriele Marschner[1], Silke Pitann[1], Sabine Adler[10], Ralf Dechend[11,12], Dominik N. Müller[11,13], Ioana Braicu[14], Jalid Sehouli[15], Kai Schulze-Forster[16,17], Tobias Trippel[18], Carmen Scheibenbogen[19,20], Annetine Staff[21], Peter R. Mertens[22], Madlen Löbel[19], Justin Mastroianni[23,24], Corinna Plattfaut[25], Frank Gieseler[25], Duska Dragun[14], Barbara Elizabeth Engelhardt[5], Maria J. Fernandez-Cabezudo[26], Hans D. Ochs[27], Basel K. al-Ramadi[28], Peter Lamprecht[1], Antje Mueller[1], Harald Heidecke[16] & Gabriela Riemekasten[1]

[1]Department of Rheumatology and Clinical Immunology, University of Lübeck, Lübeck 23538, Germany. [2]Department of Rheumatology and Clinical Immunology, Center for Chronic Immunodeficiency (CCI), Medical Center-University of Freiburg, Faculty of Medicine, University of Freiburg, Freiburg 79106, Germany. [3]Department of Statistic, Federal University of Pernambuco, Recife, PE 50670-901, Brazil. [4]Deaconess Hospital, University of Bergen, 5006 Bergen, Norway. [5]Department of Computer Science, Princeton University, Princeton, NJ 08540, USA. [6]Department of Gastroenterology, Infectiology and Rheumatology, Charité University Hospital, Berlin 12203, Germany. [7]Berlin Institute of Health (BIH), Berlin 10178, Germany. [8]Dept. of Rheumatology and Clinical Immunology, Charité University Hospital, Berlin 10117, Germany. [9]Cell Autoimmunity Group, German Rheumatism Research Center (DRFZ), Berlin 10117, Germany. [10]University Hospital and University of Bern, Bern 3012, Switzerland. [11]Experimental and Clinical Research Center, a collaboration of Max Delbruck Center for Molecular Medicine and Charité Universitätsmedizin, Berlin 13125, Germany. [12]Department of Cardiology and Nephrology, HELIOS-Klinikum Berlin, Berlin 13125, Germany. [13]Berlin Institute of Health (BIH), Berlin 10178, Germany. [14]Department of Nephrology and Cardiovascular Research, Campus Virchow, Charité University Hospital, Berlin 13353, Germany. [15]Department of Gynecology, Charité University Hospital, Berlin and Tumor Bank Ovarian Cancer Network (TOC), Berlin 13353, Germany. [16]Department of Urology, Charité University Hospital, Berlin 10117, Germany. [17]CellTrend GmbH, Luckenwalde 14943, Germany. [18]Dept. of Internal Medicine & Cardiology, Charité University Hospital, Berlin 13353, Germany. [19]Institute for Medical Immunology, Charité University Hospital Berlin, Campus Virchow, Berlin 10117, Germany. [20]Berlin-Brandenburg Center for Regenerative Therapies (BCRT), Charité University Hospital Berlin, Berlin 13353, Germany. [21]University of Oslo and Oslo University Hospital, 0372 Oslo, Norway. [22]University Clinic for Nephrology and Hypertension, Diabetes and Endocrinology, Otto-von-Guericke University Magdeburg, Magdeburg 39106, Germany. [23]Department of Hematology, Oncology and Stem Cell Transplantation, Freiburg University Medical Center, Albert Ludwigs University (ALU) of Freiburg, Freiburg 79106, Germany. [24]Faculty of Biology, Albert-Ludwigs-University (ALU), Freiburg 79104, Germany. [25]Section Experimental Oncology, University Hospital and Medical School (UKSH), University of Lübeck, Lübeck 23538, Germany. [26]Department of Biochemistry College of Medicine and Health Sciences, UAE University, Al Ain 17666, United Arab Emirates. [27]Department of Pediatrics, University of Washington School of Medicine, Seattle Children's Research Institute, Seattle, WA 98191, USA. [28]Department of Medical Microbiology and Immunology, College of Medicine and Health Sciences, UAE University, Al Ain 17666, United Arab Emirates

