## [Peer Review File · Nature Communications]

Reviewers' comments:

Reviewer #1 (Autoimmune; autoantibody)(Remarks to the Author):

The authors put forward the idea that autoantibodies are natural components of the immune system, and that they may become dysregulated, triggering the development of autoimmune diseases. This hypothesis was worked out using antibodies targeting G protein-coupled receptors. The concept was further illustrated by showing the chemotactic activity of anti-EDNRA autoantibodies.

The autoantibody interaction network was worked out using advanced statistical tools.

I have no major comments. The claims are novel and might be of interest for the immunological community.

Reviewer #2 (GPCR, signalling)(Remarks to the Author):

No specific comment for the authors.

Reviewer #3 (Flow cytometry, system biology)(Remarks to the Author):

The submitted manuscript by Cabral-Marques and colleagues on "Signatures of IgG autoantibodies targeting G protein-coupled receptors in health and disease" is an interesting and well written paper on a heavily discussed topic. I particularly appreciate the large number of patients enrolled for the various disease groups. This makes the statements credible. I have some questions and comments to the authors:

1. The authors report of presence of anti GPCR IgG in healthy and diseased individuals. They find concentration alterations of different IgGs. Concentration differences and signature changes in the signatures are related to the disease and are discussed as potential reasons for autoimmune diseases etc. For me the question arises whether the IgGs from HD and the disease groups targeting the same molecule are functionally identical for example with respect to specificity and binding affinity. Theoretically, one can imagine a situation where higher IgG concentrations could be compensated by altered binding affinity. I would like to ask the authors to expand on that.
2. My second question is going in the same direction. A higher IgG concentration in the serum could also be compensated by increased levels of exosomes carrying GPCR. Exosome concentration is elevated in various (inflammatory) diseases. Is their information that these levels are unaltered or not or did the authors even test if the concentration of exosomes carrying specific GPCRs vary in their patients?

Minor comments

1. FMO only shows up in Fig. 7A and is not explained in the text. Please add.
2. I couldn't find any information about the antibody panels used for the different cytometry assays look like. Even if commercial kits were used this should be added to the supplementary material.
3. I would like to ask the authors to add to their supplementary material also the gating scheme for the cytometry data and examples for controls.
4. No information is provided how the cytometry instrument was calibrated. This is relevant because in Fig 7 MFI values are shown and these values are dependent on the instrument sensitivity. I assume that measurements were done on a longer period of time which makes a good SOP for quality control essential. Was there any correction of the MFI values to compensate for changes in the sensitivity?
5. How was MFI ratio (Fig.7D-F) determined? Was background fluorescence subtracted for example?

6. The authors state that after the transwell assay they counted neutrophils by flow cytometry. It is not clear to me how they did that. Was there any specific staining? How did they remove cell debris from the analysis that always is produced by harvesting cells from the well? An example of a measurement would be useful.

7. Page 10 top. Here the authors state that there was no sign of apoptosis or necrosis. I couldn't find how this was tested.

We sincerely appreciate the Editor and Reviewers for their positive responses concerning our manuscript. Please find enclosed our point-by-point comments to the reviewers. In addition, we would like to acknowledge the editor and reviewers for their willingness to consider our revised manuscript for a possible publication.

REVIEWER #1.

The authors put forward the idea that autoantibodies are natural components of the immune system, and that they may become dysregulated, triggering the development of autoimmune diseases. This hypothesis was worked out using antibodies targeting G protein-coupled receptors. The concept was further illustrated by showing the chemotactic activity of anti-EDNRA autoantibodies.

The autoantibody interaction network was worked out using advanced statistical tools. I have no major comments. The claims are novel and might be of interest for the immunological community.

We appreciate **REVIEWER #1** positive response regarding the novel observations in our study. We note that there were no major or minor concerns. In addition, the reviewer found that our hypothesis was worked out with advanced statistical tools¹ and supported by experimental studies of chemotaxis. We appreciate the encouraging response.

REVIEWER #2:

No specific comment for the authors.

The reviewer seems to be satisfied with our manuscript and raised no concerns. We assume that this indicates a positive response to our study investigating the immunobiology of autoantibodies.

REVIEWER #3

The submitted manuscript by Cabral-Marques and colleagues on “Signatures of IgG autoantibodies targeting G protein-coupled receptors in health and disease” is an interesting and well written paper on a heavily discussed topic. I particularly appreciate the large number of patients enrolled for the various disease groups. This makes the statements credible. I have some questions and comments to the authors:

We are thankful to reviewer 3 who found our paper interesting and well written. He/she appreciated the large number of patients enrolled in the various disease groups making the statement credible. This referee has made constructive suggestions which we think has strengthened the revised manuscript. We provide a point-by-point response below.

1. The authors report of presence of anti GPCR IgG in healthy and diseased individuals. They find concentration alterations of different IgGs. Concentration differences and signature changes in the signatures are related to the disease and are discussed as potential reasons for autoimmune diseases etc. For me the question arises whether the IgGs from HD and the disease groups targeting the same molecule are functionally identical for example with respect to specificity and binding affinity. Theoretically, one can imagine a situation where higher IgG concentrations could be compensated by altered binding affinity. I would like to ask the authors to expand on that.

We appreciated the comment above and agree that this is of significant value to develop a more critical discussion. The topic has been included in the revised discussion (second paragraph). Here, we have determined the correlation signatures of aab in health and disease based on ELISA, an approach widely used to determine the presence of aab that target GPCRs². Although this method is well-established, there are limitations. The avidity and affinity of autoantibodies to their target are not measured. Of note, aab avidity and affinity, as well as aab isotype, can change the outcome of antibody-antigen interactions and relevant biological processes following binding. As such, some biological determinants of harboring GPCR aabs have not been measured for each aab in our study, potentially relevant for all patient groups, including patients with autoimmune diseases³⁻⁶. Consequently, it will be important to investigate the characteristics described above in HD aab to determine their influence on aab physiology. For instance, low aab concentrations could be compensated by high binding affinity, and vice versa. We are also expanding our current findings by analyzing specific epitopes. The effect of the ab is dependent on GPCR co-expression⁷, which could result in epitope spreading based on hetero-dimerization described for various GPCR. Further, we are establishing experimental immunization models to better understand the pathophysiology of aab targeting GPCRs. So far, immunizations of mice with human AGTR1 and EDNRA have been successful and will be published in detail elsewhere. For instance, immunization with AGTR1, a highly conserved receptor in humans and mice increases functional anti-AGTR1 ab levels. Such experimental immunization also induces pathological SSc features, including interstitial lung disease and skin fibrosis. Passive immune transfer studies would be a relevant next step to determine if anti-AGTR1 indeed has pathogenic effects with respect to SSc.

2. My second question is going in the same direction. A higher IgG concentration in the serum could also be compensated by increased levels of exosomes carrying GPRC. Exosome concentration is elevated in various (inflammatory) diseases. Is their information that these levels are unaltered or not or did the authors even test if the concentration of exosomes carrying specific GPRCs vary in their patients?

We have, over some time, investigated the hypothesis whether concentrations of anti-GPCR aab and their network are related to cellular expression of GPCRs and the levels of other endogenous (natural) GPCR ligands. We acknowledge the constructive suggestion made by the Reviewer and agree that exosomes could play a role in the regulation of ab levels. This is an extensive scientific problem and we have started studies last year to address the issue. However, these sizable studies warrant a separate publication. We addressed this issue in the revised discussion (Kindly see **paragraph 3**): “In addition, GPCR and growth factor receptors are not only expressed by immune cells and different body tissues, they are also present in extracellular vesicles such as exosomes, which are implicated in numerous pathologies, including cancer and inflammatory diseases⁸⁻¹⁴. Therefore, exosomes could represent another important physiological source for the stimulation of anti-GPCR aab production that remains to be investigated”

Minor comments

All flow cytometry procedures including those described in the previous questions (above) were performed as previously reported^{7,15}. We revised the method section to address all the questions raised by the reviewer. Kindly see “**Analysis of EDNRA expression by flow cytometry**”.

1. FMO only shows up in Fig. 7A and is not explained in the text. Please add.

In accordance with the recommendations from the reviewer, we explained the FMO and show an example in **Supplementary Figure 5**. Considering the multiple fluorochromes in the antibody panel to analyze EDNRA expression (**Supplementary Table 3**), the fluorescence minus one (FMO) control was determined when all the antibodies were present in the flow cytometry tube, except the antibody used to measure the EDNRA expression.

2. I couldn't find any information about the antibody panels used for the different cytometry assays look like. Even if commercial kits were used this should be added to the supplementary material.

Antibody panel used for the flow cytometric analyses of EDNRA expression is now shown in a new table (**Supplementary Table 3**).

3. I would like to ask the authors to add to their supplementary material also the gating scheme for the cytometry data and examples for controls.

An example of gating scheme for the flow cytometry analyses and examples were provided in **Supplementary Figure 5**.

4. No information is provided how the cytometry instrument was calibrated. This is relevant because in Fig 7 MFI values are shown and these values are dependent on the instrument sensitivity. I assume that measurements were done on a longer period of time which makes a good SOP for quality control essential. Was there any correction of the MFI values to compensate for changes in the sensitivity?

We provided the information (please refer to **Analysis of EDNRA expression by flow cytometry**) CytoFLEX and MACS Quant Flow cytometers were calibrated using Cytoflex Daily QC (Beckman Coulter) and MACSQuant™ Calibration Beads (Miltenyi Biotec), respectively, according to the manufacturers' instructions. For MFI values an isotype control (**Supplementary Table 3**) was used as described below to compensate for changes in the cytometry instrument sensitivity.

5. How was MFI ratio (Fig.7D-F) determined? Was background fluorescence subtracted for example?

We addressed this issue in the method section (kindly see: **Analysis of EDNRA expression by flow cytometry**). MFI ratio was determined by controlling the daily technical variability using a fluorochrome-conjugated isotype control antibody. A subtraction procedure¹⁶ was applied to obtain the EDNRA MFI density as calculated below.

$$\text{EDNRA density} = \frac{\text{MFI}_{\text{ETAR}}}{\text{MFI}_{\text{Isotype}}}$$

6. The authors state that after the transwell assay they counted neutrophils by flow cytometry. It is not clear to me how they did that. Was there any specific staining? How did they remove cell debris from the analysis that always is produced by harvesting cells from the well? An example of a measurement would be useful.

We have clarified this point in the method section (kindly see **Migration Assays**). For the analysis of HD-IgG-induced migration, neutrophils were isolated from heparinized blood by dextran sedimentation. This was followed by Ficoll-Hypaque centrifugation, as previously described¹⁷. Purity, based on CD15 expression, was always above 97%, as was viability before migration assays. A fluorescence microscope (EVOS FL Cell Imaging System, Oakwood, OH, USA) was used to confirm the presence of neutrophils at the bottom surface of transwell plates. Neutrophils present in the bottom of the transwell plates after migration (**Fig. 7B, lower panel**) were transferred to a 96-well plate and counted (cells/μl) by flow cytometry. To exclude cell debris, gating was based on neutrophil size (forward scatter, FSC) and granularity (side scatter, SSC). In addition, the cell migration toward HD-IgG in an EDNRA-dependent manner was confirmed using a Cell based Oris™ migration assay with Colo357 cells, which is currently shown as an example and described in **Supplementary Figure 6**.

7. Page 10 top. Here the authors state that there was no sign of apoptosis or necrosis. I couldn't find how this was tested.

We have made this clearer and provided an additional reference¹⁸. Kindly see lines 10-11 (**Chemotactic activity of anti-EDNRA aab**): Apoptosis and necrosis were tested as shown, described, and exemplified in **Supplementary Figure 7**.

REFERENCES

1. Genser, B., Cooper, P. J., Yazdanbaksh, M., Barreto, M. L. & Rodrigues, L. C. A guide to modern statistical analysis of immunological data. *BMC Immunol.* **8**, 27 (2007).
2. Cabral-Marques, O. & Riemekasten, G. Functional autoantibodies targeting G protein-coupled receptors in rheumatic diseases. *Nat. Rev. Rheumatol.* **13**, 648–656 (2017).
3. Meyer, S. *et al.* AIRE-Deficient Patients Harbor Unique High-Affinity Disease-Ameliorating Autoantibodies. *Cell* **166**, 582–595 (2016).
4. Bordon, Y. Autoantibodies with a silver lining? *Nat. Rev. Immunol.* **16**, 536–536 (2016).
5. McHugh, J. Hormone status regulates autoantibody pathogenicity. *Nat. Rev. Rheumatol.* **14**, 385–385 (2018).
6. Elkon, K. & Casali, P. Nature and functions of autoantibodies. *Nat. Clin. Pract. Rheumatol.* **4**, 491–8 (2008).
7. Rademacher, J. *et al.* Monocytic Angiotensin and Endothelin Receptor Imbalance Modulate Secretion of the Profibrotic Chemokine Ligand 18. *J. Rheumatol.* **43**, 587–91 (2016).
8. Fyfe, I. Exosomes can spread toxic AD pathology. *Nat. Rev. Neurol.* **14**, 451–451 (2018).
9. Wermuth, P. J., Piera-Velazquez, S. & Jimenez, S. A. Exosomes isolated from serum of systemic sclerosis patients display alterations in their content of profibrotic and antifibrotic microRNA and induce a profibrotic phenotype in cultured normal dermal fibroblasts. *Clin. Exp. Rheumatol.* **35 Suppl 106**, 21–30
10. Nager, A. R. *et al.* An Actin Network Dispatches Ciliary GPCRs into Extracellular Vesicles to Modulate Signaling. *Cell* **168**, 252–263.e14 (2017).
11. Németh, A. *et al.* Antibiotic-induced release of small extracellular vesicles (exosomes) with surface-associated DNA. *Sci. Rep.* **7**, 8202 (2017).
12. Heath, N. *et al.* Rapid isolation and enrichment of extracellular vesicle preparations using anion exchange chromatography. *Sci. Rep.* **8**, 5730 (2018).
13. Isola, A. L. & Chen, S. Exosomes: The Link between GPCR Activation and Metastatic Potential? *Front. Genet.* **7**, 56 (2016).
14. Gamperl, H. *et al.* Extracellular vesicles from malignant effusions induce tumor cell migration: inhibitory effect of LMWH tinzaparin. *Cell Biol. Int.* **40**, 1050–1061 (2016).
15. Günther, J. *et al.* Angiotensin receptor type 1 and endothelin receptor type A on immune cells mediate migration and the expression of IL-8 and CCL18 when stimulated by autoantibodies from systemic sclerosis patients. *Arthritis Res. Ther.* **16**, R65 (2014).
16. Overton, W. R. Modified histogram subtraction technique for analysis of flow cytometry data. *Cytometry* **9**, 619–626 (1988).
17. Cabral-Marques, O. *et al.* CD40 ligand deficiency causes functional defects of

peripheral neutrophils that are improved by exogenous IFN- γ . *J. Allergy Clin. Immunol.* (2018). doi:10.1016/j.jaci.2018.02.026

18. Martin, S. J. *et al.* Early redistribution of plasma membrane phosphatidylserine is a general feature of apoptosis regardless of the initiating stimulus: inhibition by overexpression of Bcl-2 and Abl. *J. Exp. Med.* **182**, 1545–1556 (1995).

REVIEWERS' COMMENTS:

Reviewer #3 (Remarks to the Author):

The authors have answered all my questions and concerns and made appropriate editions to their manuscript. I have no further questions or comments and thank the authors.

REVIEWERS' COMMENTS:

Reviewer #3 (Remarks to the Author):

The authors have answered all my questions and concerns and made appropriate editions to their manuscript. I have no further questions or comments and thank the authors.

We acknowledge the **Reviewer #3** for his positive response. The referee has made some constructive suggestions which we think has strengthened our manuscript